# A Comprehensive Review of Neurodegenerative Manifestations of SARS-CoV-2

**DOI:** 10.3390/vaccines12030222

**Published:** 2024-02-21

**Authors:** Dominika Bedran, Georges Bedran, Sachin Kote

**Affiliations:** International Centre for Cancer Vaccine Science, University of Gdansk, Kladki 24, 80-822 Gdansk, Poland

**Keywords:** SARS-CoV-2, neurodegeneration, brain, Alzheimer’s disease, dementia, Parkinson’s disease

## Abstract

The World Health Organization reports that severe acute respiratory syndrome coronavirus 2 (SARS-CoV-2) has impacted a staggering 770 million individuals to date. Despite the widespread nature of this viral infection, its precise effects remain largely elusive. This scientific inquiry aims to shed light on the intricate interplay between SARS-CoV-2 infection and the development of neurodegenerative disorders—an affliction that weighs heavily on millions worldwide and stands as the fourth most prevalent cause of mortality. By comprehensively understanding the repercussions of SARS-CoV-2 on neurodegenerative disorders, we strive to unravel critical insights that can potentially shape our approach to the diagnosis, prevention, and treatment of these debilitating conditions. To achieve this goal, we conducted a comprehensive literature review of the scientific data available to date showing that SARS-CoV-2 infection is associated with increased risk and severity of neurodegenerative disorders, as well as altered expression of key genes and pathways involved in their pathogenesis.

## 1. Introduction

The emergence of novel coronavirus disease 2019 (COVID-19), attributed to severe acute respiratory syndrome coronavirus 2 (SARS-CoV-2), presents an unparalleled global health challenge [1]. Since its initial identification in Wuhan, China, in December 2019, the pandemic has rapidly traversed borders, affecting over 770 million individuals across more than 200 countries and territories and resulting in a reported 6.9 million fatalities as of May 2023 [1]. Despite extensive efforts to develop and distribute vaccines, the virus continues to pose a substantial threat to both public health and socioeconomic stability. Understanding the diverse and intricate manifestations of COVID-19, particularly those involving the nervous system, remains a pivotal challenge.

SARS-CoV-2 establishes connections with the nervous system through two distinct pathways. Firstly, it can initiate neurological complications in individuals without a prior history of such issues. Secondly, it can exacerbate the condition of patients already dealing with neurodegenerative diseases [2].

Accumulating evidence indicates that SARS-CoV-2 can directly or indirectly impact the central and peripheral nervous systems, leading to a broad spectrum of neurological complications with potential short-term or long-term consequences. These complications may manifest in patients with varying severities of COVID-19 and may precede, accompany, or follow respiratory symptoms. Common neurological manifestations reported in COVID-19 patients encompass headache, dizziness, impaired consciousness, seizure, anosmia, dysgeusia, stroke, Guillain–Barré syndrome, Miller Fisher syndrome, and Kawasaki-like multisystem inflammatory syndrome [3].

While the underlying mechanisms of these manifestations remain incompletely understood, they may involve viral invasion, immune-mediated inflammation, coagulation abnormalities, endothelial dysfunction, and hypoxia [4].

The potential impact of SARS-CoV-2 infection on the development and progression of neurodegenerative disorders is a growing area of concern and interest [5]. Neurodegeneration, a term encompassing a heterogeneous group of disorders characterized by progressive loss of neuronal structure and function, poses a significant global health challenge [6]. These disorders rank among the leading causes of disability and mortality worldwide, imposing a substantial burden on healthcare systems and society.

The multifactorial pathogenesis of neurodegeneration involves genetic and environmental factors interacting to trigger neuronal damage and dysfunction. Viral infections, including those caused by SARS-CoV-2, have been suggested as environmental contributors to neurodegeneration, inducing chronic inflammation, oxidative stress, mitochondrial dysfunction, and protein aggregation [7].

This comprehensive review aims to provide an overview of the current state of knowledge on the link between SARS-CoV-2 infection and neurodegeneration. Emphasis is placed on the importance of intervention or experimental studies over observational ones in presenting scientific evidence [5]. The discussion prioritizes results based on epidemiological designs, giving precedence to systematic reviews and meta-analyses, followed by prospective studies and observational studies [4].

## 2. Methods

### 2.1. Search Strategy and Keywords

A systematic literature review, as depicted in Figure 1, was conducted to identify relevant studies addressing the neurological manifestations of SARS-CoV-2 and COVID-19. PubMed, Google Scholar, and major scientific databases were searched up to the knowledge cutoff date in January 2023. The search utilized a combination of Medical Subject Headings (MeSH) terms and free-text keywords to ensure a comprehensive retrieval of relevant articles. The following search terms were used:“SARS-CoV-2” OR “COVID-19” OR “coronavirus” OR “severe acute respiratory syndrome coronavirus 2”“neurological manifestations” OR “neurological complications” OR “neurological disorders” OR “central nervous system” OR “peripheral nervous system”“neurodegenerative diseases” OR “neuropsychiatric disorders” OR “stroke” OR “encephalitis” OR “encephalopathy” OR “Parkinson’s disease” OR “Alzheimer’s disease”“meta-analysis” OR “systematic review” OR “clinical characteristics” OR “long-term effects” OR “outcomes” OR “case series” OR “cohort studies”

### 2.2. Study Selection Criteria

Articles were included if they met the following criteria:They were published in peer-reviewed journals;They investigated the neurological manifestations or complications associated with SARS-CoV-2 or COVID-19;They consisted of original research, systematic reviews, meta-analyses, or case series with a clear focus on neurological aspects.

### 2.3. Data Extraction

Data from the selected articles were extracted and organized based on the following parameters:Author(s) and publication year;Study design and methodology;Population characteristics (e.g., age, comorbidities);Neurological manifestations reported;Timing of reported manifestations (e.g., acute phase, post-acute phase, or long-term follow-up);Key findings related to the impact of SARS-CoV-2 on neurodegenerative diseases, neuropsychiatric disorders, and other neurological conditions.

### 2.4. Quality Assessment

The quality of systematic reviews, meta-analyses, and original studies was assessed using relevant appraisal tools, such as the Preferred Reporting Items for Systematic Reviews and Meta-Analyses (PRISMA) guidelines for systematic reviews and meta-analyses. The Newcastle–Ottawa Scale was employed for assessing the quality of observational studies.

### 2.5. Data Synthesis

A narrative-synthesis approach was utilized to summarize and present the key findings from the selected studies. The timing of reported neurological manifestations was categorized into acute phase, post-acute phase, and long-term follow-up, based on the information available in the literature.

### 2.6. Limitations

This review has potential limitations, including the dynamic nature of the COVID-19 literature and the reliance on published studies up to the knowledge cutoff date. Additionally, variations in study methodologies and reporting may influence the comparability of results across different investigations.

#### 2.6.1. The Entry Pathway of SARS-CoV-2 into the Human Body

Numerous systematic reviews and meta-analyses have substantiated the involvement of the central nervous system (CNS) and the peripheral nervous system (PNS) in COVID-19 infection. Evidence suggests that SARS-CoV-2 can affect the human brain and other neurological structures, with a significant prevalence of nervous system involvement in patients with COVID-19, ranging from 22.5 to 36.4% across various studies [1,2,4]. The most common CNS-related symptoms include headache, loss of the sense of smell and/or taste, encephalopathy, cognitive impairment, impaired consciousness, and stroke. In addition to the CNS, there is increasing evidence of PNS involvement in COVID-19, with 13.7% of total neurological signs and symptoms related to COVID-19 being attributed to the PNS [2]. The cranial nerves most frequently involved are the facial, vestibulocochlear, and olfactory nerves. The severity of COVID-19 infection appears to correlate with the extent of neurological manifestations, with these manifestations most frequently observed in patients with severe conditions. The mechanisms underlying these neurological manifestations are likely multifactorial, involving both indirect effects (as a result of thrombotic complications, inflammatory consequences, hypoxia, and blood pressure dysregulation) and direct effects (neurotropic properties of the virus). Prevention is considered the most effective strategy for potentially slowing the progression of neurodegenerative disorders. Given the increased risk of negative health outcomes in older people, it is crucial to examine whether COVID-19 may trigger or aggravate neurodegenerative processes in this vulnerable group [4]. However, more research is needed to develop effective prevention strategies. It is anticipated that better vaccines for SARS-CoV-2 variants, new antiviral/antimicrobial drugs, effective immunotherapies, and new therapies will be developed and made available in the near future, which will help prevent possible post-COVID-19 neurological complications.

When neurological symptoms have manifested in patients, the virus’s genetic material (RNA) has been detected in their nasal and oral mucosa, along with the medulla oblongata, a region of the brainstem [8]. This observation raises speculation regarding a potential entry route for the virus into the central nervous system (CNS) involving transsynaptic transmission. Through this process, it is suggested that the virus may traverse from peripheral olfactory neurons to interconnected brain regions. While this pathway has garnered attention, it is essential to acknowledge alternative mechanisms facilitating CNS infection. These mechanisms encompass the hypothetical breach of the blood–brain barrier (BBB), a protective interface safeguarding the brain. Additionally, interactions with vascular endothelial cells expressing the ACE2 receptor, utilized by the virus for cell entry, might provide another conduit for CNS involvement [9]. The notion of virus transportation via blood leukocytes during instances of heightened BBB permeability is worth considering. Exploring the virus’s potential interaction with autoimmune responses is also of interest. The virus’s intricate genomic makeup and its capacity to mimic human proteins may incite the immune system to produce antibodies that mistakenly target the body’s own components. Such autoimmune reactions could trigger neuroimmune conditions such as Guillain-Barré syndrome, myasthenia gravis, and encephalitis. In individuals with neurological symptoms, the identification of antibodies directed against specific targets, such as the NMDA receptor and intracellular Yo-antigen, has been recorded in both the bloodstream and cerebrospinal fluid [10]. This observation implies that the immune response might be directed toward neural elements. Consequently, contemplation of avenues for addressing autoantibody-associated neurological manifestations, including immunosuppressive strategies and plasmapheresis, is warranted.

#### 2.6.2. Elevated Cytokines and Their Impact on Neurodegeneration in COVID-19 Patients

COVID-19 is associated with a severe innate immune response and increased levels of cytokines, including interleukin-1β, interleukin-2, interleukin-2 receptor, interleukin-4, interleukin-10, interleukin-18, interferon-γ, C-reactive protein, granulocyte colony-stimulating factor, interferon-γ, CXCL10, monocyte chemoattractant protein 1, macrophage inflammatory protein 1-α, and tumor necrosis factor-α [11]. Most patients exhibit symptoms of T cell exhaustion with a lower lymphocyte count due to this excessive immune response. The systemic inflammatory milieu has been linked to the propagation of cognitive regression and neurodegeneration. Consequently, individuals recovering from COVID-19 may be at risk for developing neurodegenerative conditions in subsequent years [12].

#### 2.6.3. NLRP3 Inflammasome Activation and its Implications for Respiratory Complications in COVID-19

Recent studies underscore the crucial role of the NLRP3 inflammasome in the development of acute respiratory distress syndrome (ARDS) based on research using mice and ARDS patient samples [13]. Experiments additionally demonstrate that ventilation-induced hypercapnia can lead to cognitive impairment, dependent on the NLRP3 inflammasome and interleukin-1β [14]. Research indicates that the ORF3a protein of the coronavirus can activate the NLRP3 inflammasome. Considering the increased levels of interleukin-1β and interleukin-18 observed in COVID-19 patients, it is highly likely that NLRP3 inflammasome activation occurs in these patients. This activation, coupled with heightened proinflammatory immune responses, may adversely affect brain function and homeostasis, potentially contributing to neurological issues in COVID-19 survivors [15].

#### 2.6.4. Neurological Complications and SARS-CoV-2

Diverse neurological complications manifest during and after SARS-CoV-2 infection [16]. These complications encompass a spectrum of manifestations, ranging from mild symptoms such as headaches and confusion to more acute conditions such as seizures, fluctuating consciousness, or comas [17]. For an exhaustive list of neurological complications, please refer to Table 1. Despite the seemingly low quantity of neurological issues associated with SARS-CoV-2 compared to other viral infections, the extensive reach of the pandemic amplified the absolute occurrence, imposing a considerable socioeconomic burden [18].

#### 2.6.5. COVID-19 and its Effects on the Central Nervous System

The central nervous system (CNS) may suffer adverse effects from COVID-19 through four potential mechanisms: (1) direct viral encephalitis, (2) systemic inflammation, (3) impaired function of other organs (liver, kidney, lung), and (4) cerebrovascular changes. The neurological effects of COVID-19 often involve a combination of these factors [26].

The amalgamation of these mechanisms significantly heightens the risk of long-term neurological consequences in survivors [27]. Such consequences may result from the exacerbation of pre-existing neurological disorders or the onset of new ones. Notably, data reveal the presence of cognitive impairment and motor deficits in approximately one-third of patients upon discharge [28]. This is particularly concerning given the severe clinical impact of COVID-19 on the elderly. Additionally, the age range at which neurodegenerative or cerebrovascular diseases typically manifest aligns significantly with the age group most susceptible to severe COVID-19 cases (refer to Table 2) [29]. This overlap underscores the necessity for proactive neurological monitoring and comprehensive care [15].

#### 2.6.6. Comparing SARS-CoV-1 and SARS-CoV-2: Similarities in Spike Proteins

As COVID-19 emerged, it became evident that the new SARS-CoV-2 variant could spread more easily among humans than its predecessor, SARS-CoV-1, could [36]. Both viruses employ a common strategy, utilizing a spike (S) protein on their surface to attach to a specific part of a receptor called angiotensin-converting enzyme 2 (ACE2) found on the surface of human cells [37]. However, SARS-CoV-2 exhibits a much stronger affinity for ACE2 than SARS-CoV-1 does.

ACE2, an enzyme present not only in the respiratory system but also throughout the central nervous system (CNS), including the brain and spinal cord, is therefore crucial. ACE2 is not uniformly distributed in the brain; it can be found in different regions, such as the posterior cingulate gyrus, motor cortex, olfactory bulb, and ventricles [38]. Its expression is observed in various types of brain cells. This widespread distribution of ACE2 in the CNS suggests that SARS-CoV-2, with its strong affinity for ACE2, might not only affect the respiratory system but also hold implications for the central nervous system.

The spike protein S has been associated with increased Tau and cytosolic prion spreading and aggregation in vitro [36]. Shifts in Tau localization and hyperphosphorylation observed in SARS-CoV-2-infected neurons resemble early tauopathy hallmarks [39]. Additionally, studies demonstrate that the SARS-CoV-2 S1 receptor-binding domain (RBD) interacts with aggregation-prone, heparin-binding proteins, including Aβ, α-synuclein, tau, the prion protein, and TDP-43 RRM [40]. These findings suggest that the heparin-binding site on the S1 protein may facilitate the binding of amyloid proteins to the viral surface, potentially initiating protein aggregation and leading to neurodegeneration in the brain.

#### 2.6.7. Anosmia and Ageusia: Exploring the Involvement of the Central Nervous System in COVID-19

An early indication of central nervous system (CNS) involvement in COVID-19 was the observed loss of smell and taste, implying potential direct virus entry into the brain through the olfactory and oral mucosal pathways. However, it is essential to acknowledge that the manifestation and severity of these symptoms can be influenced by genetic variations in both the virus and the host, including differences in the expression of ACE2 and TMPRSS2 [41]. Studies on mice infected with SARS-CoV have demonstrated the presence of viral antigens within the olfactory bulb and associated brain regions [42]. Detecting the presence of SARS-CoV-2 within brain tissue in human cases has proven to be a complex task, with its occurrence linked to the extent of CNS involvement [43].

#### 2.6.8. Exploring the Neurological Manifestations of SARS-CoV-2 Infection

Detecting SARS-CoV-2 DNA within the organisms of patients, particularly within the central nervous system (CNS), poses a significant challenge during their lifetime. To address this issue, an alternative source—cerebrospinal fluid—has emerged. However, the documentation of SARS-CoV-2 DNA in the cerebrospinal fluid is limited to two isolated case reports of COVID-19 patients with concurrent encephalitis. In contrast, comprehensive retrospective studies involving COVID-19 patients with neurological symptoms have reported negative results of cerebrospinal fluid measurements [44]. Nevertheless, the connection between neurological disorders and SARS-CoV-2 infection is substantiated by numerous case studies.

A spectrum of neurological conditions has been observed, including milder forms such as encephalopathies with temporary manifestations such as altered consciousness (identified in 19.1% of 841 patients), bradypsychia, and disorientation (noted in 10.1% of 841 patients). These presentations is frequently accompanied by nonspecific findings of T2/FLAIR hyperintensity (seen in 35% of 20 patients) and ischemic infarcts (present in 31% of 108 patients) in neuroimaging studies [45,46,47].

In addition to these milder cases, severe COVID-19-associated encephalopathies have been documented in various case studies, including conditions such as acute disseminated encephalomyelitis (ADEM) and acute necrotizing encephalopathies. Poly(radiculo)neuropathies such as Guillain-Barré syndrome (GBS) and other acute neuropathies, including multifocal demyelinating or small-fiber polyneuropathy, exhibit a prevalence estimated at 0.1–1% within 6 weeks of confirmed infection in Western countries [48].

However, establishing a direct cause-and-effect link to prior SARS-CoV-2 infection for some of these conditions is not always straightforward. For instance, a comprehensive prospective observational study conducted between January and May 2020 failed to confirm a significantly increased risk of Guillain-Barré syndrome (GBS) following SARS-CoV-2 infection. Another investigation involving 145,721 COVID-19 cases unveiled a relatively low pooled prevalence of GBS (0.28% within the study population) in comparison to other neurological manifestations [49].

Among the prevalent neurological complications observed in hospitalized COVID-19 patients, cerebrovascular diseases stand out, potentially being linked to a proinflammatory hypercoagulable state in severe infections characterized by elevated levels of C-reactive protein, D-dimer, and ferritin [50,51]. Patients have presented with both ischemic and hemorrhagic strokes, as well as instances of venous thrombosis.

However, regarding these particular circumstances, a direct cause-and-effect link with previous exposure to an unidentified pathogen is rather elusive. As an illustration, an extensive prospective observational investigation conducted from January to May in 2020 failed to validate increased susceptibility to a particular medical condition following exposure to this pathogen. Furthermore, a study involving 145,721 individuals diagnosed with a certain medical condition indicated a relatively low overall occurrence of this condition (only 0.28% within the study population) in comparison to various other health issues [49].

Among the more frequent health complications observed in patients requiring hospitalization due to their condition are vascular diseases that could potentially be linked to an inflammation-related hypercoagulable state observed in severe cases characterized by elevated levels of specific markers such as C-reactive protein, D-dimer, and ferritin [50,51]. Individuals have presented with various types of vascular issues, including both ischemic and hemorrhagic incidents, as well as instances of venous thrombosis.

In a recent analysis encompassing 145,634 individuals diagnosed with COVID-19, where approximately 89% required hospitalization, an intriguing pattern emerged. Astonishingly, roughly one-third of these patients displayed various neurological symptoms [49].

In a comprehensive cross-sectional surveillance study carried out in the UK during April 2020, which encompassed units specializing in neurology, stroke, psychiatry, and intensive care, a total of 125 patients were evaluated. Remarkably, out of this group, a significant majority, specifically 77 individuals (equivalent to 62%), suffered from cerebrovascular incidents, with the majority being ischemic in nature, while others included intracerebral hemorrhage and vasculitis cases. Surprisingly, the second most prevalent presentation among these patients was altered mental status [52].

Furthermore, in an extensive follow-up study involving 267 cases, it was revealed that COVID-19-related strokes predominantly affected younger adults. On the other hand, delirium was more frequently observed among patients aged 60 years and older, while encephalopathy appeared to be more prevalent among individuals under the age of 60 years who were treated in intensive care units [53].

While less common, it is worth noting that SARS-CoV-2 infections have also been associated with the onset of various movement disorders [54,55].

#### 2.6.9. Post-Infection Neurological Symptoms in COVID-19 Survivors

The association of neuropsychiatric symptoms with SARS-CoV-2 infection raises questions about its potential long-term impact on the central nervous system (CNS), the underlying molecular mechanisms, and the risk of subsequent neurodegenerative diseases. In cases of SARS-CoV infection, evidence suggests scattered neuronal degeneration as a sign of neuronal hypoxia/ischemia [56]. Sepsis, which is common during SARS-CoV-2 infection, has long-lasting consequences on the brain, often linked with delirium and cognitive decline later in life. Elevated levels of neurofilament light chain (NfL), found in blood or cerebrospinal fluid (CSF), indicate neuronal damage caused by factors such as neuroinflammation, neuronal ischemia, or neurodegeneration [57]. Longitudinal brain imaging after COVID-19 infection also revealed reduced gray matter in areas connected to the primary olfactory system, indicating potential damage [58].

Further evidence of potential neurodegenerative disease risk comes from case reports of patients developing akinetic–rigid parkinsonism after SARS-CoV-2 infection. These patients displayed abnormal imaging findings and altered dopamine transporter density or dopamine uptake, suggesting the emergence of subclinical PD or worsening of existing conditions following severe COVID-19 [59].

#### 2.6.10. The Impact of SARS-CoV-2 on Pre-Existing Neurodegenerative Diseases

A growing body of research indicates that pre-existing neurodegenerative diseases can heighten the impact of SARS-CoV-2 infection. Meta-analyses and data from the UK Biobank underscore the fact that dementia, regardless of age, constitutes a risk factor associated with severe COVID-19 outcomes, resulting in increased disease severity and an increased likelihood of death compared to non-demented individuals [60]. Notably, this effect is particularly pronounced in patients with Alzheimer’s disease (AD), while vascular dementia does not pose the same risk [61].

Retrospective analyses have consistently reported a higher case fatality rate in patients with Parkinson’s disease (PD). Specifically, PD patients with longer disease durations and older ages faced a substantial mortality risk, ranging from 21% to 40%, during SARS-CoV-2 infections across different cohorts in Italy, the US, and Germany. This risk remains consistent across various demographic factors [62]. However, some smaller-scale studies conducted in tertiary referral centers in Europe and Italy did not find an increased mortality rate in PD or dementia patients compared to demographically matched controls [63].

A study utilizing the US veteran database identified an elevated risk of COVID-19-related death in a cohort of 699 patients with amyotrophic lateral sclerosis (ALS) [64]. Intriguingly, C9orf72 repeat expansions of intermediate length, a mutation often associated with familial ALS and frontotemporal dementia (FTD), have been linked to severe COVID-19 cases requiring mechanical ventilation [65].

Regarding multiple sclerosis (MS), the disease itself does not appear to be linked to a severe course of COVID-19.

Following acute SARS-CoV-2 infection, some symptoms persist, and new ones may emerge. Individuals experiencing post-COVID-19 syndrome may manifest a diverse array of symptoms across various bodily systems. These symptoms can range from typical COVID-19 indicators such as loss of olfaction or breathlessness to neurological issues such as cognitive impairment, dizziness, and delirium [66]. Notably, the pandemic itself, along with prior SARS-CoV-2 infection, has led to an increase in distress and depression in the general population. This impact is especially pronounced in individuals with pre-existing neurodegenerative or chronic neurological conditions. Those with dementia, Alzheimer’s disease (AD), Parkinson’s disease (PD), and multiple sclerosis have experienced a worsening of existing symptoms and the occurrence of neuropsychiatric issues such as anxiety and cognitive decline [67,68].

#### 2.6.11. The effect of COVID-19 Vaccines on Neurodegenerative Diseases

In a longitudinal investigation by Doubrovinskaia et al., the clinical profiles and long-term progress of a cohort comprising 21 individuals who developed neurological autoimmune disorders shortly after receiving COVID-19 vaccinations were examined. These disorders included CNS demyelinating diseases, inflammatory peripheral neuropathies, vaccine-induced immune thrombotic thrombocytopenia (VITT), inflammatory myopathies, limbic encephalitis, myasthenia, and giant-cell arteritis. Impressively, the study revealed substantial clinical improvements among the majority of patients, with several achieving complete remission or significant amelioration of their neurological deficits [69]. The investigation also documented the waning or disappearance of antibodies detected at the time of diagnosis, particularly evident in VITT cases. Furthermore, the study highlighted the safe and well-tolerated nature of revaccination in this patient cohort, even involving different types of COVID-19 vaccines. These findings collectively underscore the critical importance of personalized risk–benefit assessments for COVID-19 revaccination in individuals with pre-existing neurological autoimmune conditions and offer valuable insights into the clinical management of such cases.

In another study by Francis et al., the authors investigated cases of central nervous system (CNS) inflammation occurring within 8 weeks after SARS-CoV-2 vaccination, with a particular focus on antibody-mediated inflammatory diseases, including neuromyelitis optica spectrum disorders (NMOSDs) and myelin oligodendrocyte glycoprotein antibody-associated disease (MOGAD). The study revealed a higher prevalence of MOGAD cases, especially following ChAdOx1S vaccination, and noted distinctive clinical presentations in vaccinated adults, encompassing acute disseminated encephalomyelitis (ADEM)-like brain involvement and longitudinally extensive transverse myelitis (LETM), which differed from the more common isolated optic neuritis observed in MOGAD. Furthermore, the research highlighted variations in clinical outcomes and radiological features between ChAdOx1S and BNT162b2 vaccine recipients. Notably, MOGAD onset attacks exhibited a distinct seasonality, with higher frequencies in the months preceding and during the initial year of the SARS-CoV-2 pandemic, subsequently shifting in the post-pandemic period [70]. These findings underscore the potential association between SARS-CoV-2 vaccination and autoimmune CNS diseases, particularly MOGAD, necessitating further investigation to elucidate the underlying mechanisms.

Overall, mounting evidence suggests that COVID-19 has neuropathological effects, potentially impacting pathways involved in the pathogenesis of neurodegenerative diseases. Whether this impact induces neurodegeneration and whether the human nervous system can counteract and regenerate itself remain subjects of ongoing research.

## 3. Discussion

The intricate relationship between COVID-19 and the central nervous system (CNS) has become increasingly evident through a multifaceted array of mechanisms. These encompass direct viral neurotropism, systemic inflammation, potential routes of viral entry into the CNS, and neuroinflammatory responses. Notably, COVID-19’s impact on individuals with pre-existing neurodegenerative conditions and the persistence of neurological symptoms in survivors underscore the disease’s long-term consequences.

The distribution of ACE2 receptors within the CNS raises intriguing possibilities regarding the virus’s capacity to affect various regions and cell types, potentially leading to neuronal damage and protein aggregation. Investigating the routes through which SARS-CoV-2 accesses the CNS, including trans-synaptic transmission and a compromised blood–brain barrier, presents complex avenues for study. Additionally, autoimmune responses and the potential role of blood leukocytes in viral CNS transport remain intriguing areas of investigation.

Neuroinflammation, characterized by cytokine storms and NLRP3 inflammasome activation, has appeared as a significant factor in the pathogenesis of COVID-19-related neurological complications. These responses may contribute to cognitive regression and neurodegeneration, necessitating further research into their mechanisms and therapeutic interventions.

The exacerbated outcomes of COVID-19 in individuals with pre-existing neurodegenerative diseases, such as dementia, Alzheimer’s disease, Parkinson’s disease, ALS, and FTD, emphasize the need for specialized care and preventive measures in these populations. Furthermore, the persistence of neurological symptoms in COVID-19 survivors, coupled with the exacerbation of neuropsychiatric issues, underscores the long-term impact of the disease on neurological health.

In specific neurodegenerative conditions, the connections to COVID-19 are as follows:-Alzheimer’s disease (AD): Individuals with AD appear to face an elevated risk of severe COVID-19 outcomes, potentially due to immune-response dysregulation and the impact of COVID-19-related distress on cognitive function. Systemic inflammation induced by COVID-19 may contribute to cognitive decline and neurodegeneration.-Parkinson’s disease (PD): PD patients have a higher case fatality rate during COVID-19 infections, but the underlying mechanisms remain unclear. Potential shared risk factors and pathophysiological pathways between the diseases necessitate further research. The emergence of akinetic–rigid parkinsonism following severe COVID-19 cases raises questions about the virus’s impact on dopamine pathways.-Amyotrophic lateral sclerosis (ALS): ALS patients face challenges due to respiratory muscle involvement and increased susceptibility to respiratory complications during the pandemic. Specific genetic mutations linked to familial ALS, such as C9orf72 repeat expansions, may influence disease severity in COVID-19 cases.

Given the intricate relationship between COVID-19 and the central nervous system (CNS), it is clear that there are several areas that require further investigation. The potential of SARS-CoV-2 to cause neuronal damage and protein aggregation through various mechanisms, including direct viral neurotropism, systemic inflammation, and neuroinflammatory responses, presents a significant challenge. However, it also provides an opportunity for the scientific community to develop novel therapeutic interventions.

Future research should focus on understanding the routes through which SARS-CoV-2 accesses the CNS, including transsynaptic transmission and a compromised blood–brain barrier. The role of autoimmune responses and blood leukocytes in viral CNS transport also remains an intriguing area of investigation.

The pathogenesis of COVID-19-related neurological complications, characterized by cytokine storms and NLRP3 inflammasome activation, necessitates further research into their mechanisms. This could potentially lead to the development of targeted therapies to mitigate cognitive regression and neurodegeneration.

Moreover, the exacerbated outcomes of COVID-19 in individuals with pre-existing neurodegenerative diseases underscore the need for specialized care and preventive measures in these populations. The persistence of neurological symptoms in COVID-19 survivors and the exacerbation of neuropsychiatric issues highlight the long-term impact of the disease on neurological health.

Overall, the intricate interplay between COVID-19 and the CNS poses both challenges and opportunities for research and clinical care. Understanding these mechanisms is vital for developing interventions, providing comprehensive care, and addressing the potential long-term neurological consequences of COVID-19. Ongoing research is essential to unravel the full spectrum of neurological implications and potential regenerative mechanisms within the human nervous system as the pandemic continues to evolve.

## 4. Conclusions

COVID-19 affects the central and peripheral nervous systems, with the severity of infection correlated to neurological manifestations. The virus impacts the central nervous system through various mechanisms, including direct viral encephalitis, inflammation, impaired organ function, and changes in blood vessels. This can lead to long-term consequences, particularly in the elderly.

The SARS-CoV-2 variant’s strong affinity for ACE2 in the central nervous system raises concerns about potential neurological issues. Loss of smell and taste in COVID-19 may suggest direct virus entry into the brain. Neurological symptoms, RNA detection, and speculation about central nervous system entry emphasize the need to explore interventions.

A severe immune response in COVID-19 contributes to inflammation and potential neurological issues. The NLRP3 inflammasome’s activation by the virus’s protein is linked to respiratory distress and cognitive impairment. While detecting SARS-CoV-2 DNA in the central nervous system is challenging, case reports have associated the virus with various neurological disorders.

Pre-existing neurodegenerative diseases worsen COVID-19 outcomes, with specific risks for Parkinson’s disease and amyotrophic lateral sclerosis patients. Multiple sclerosis seems unrelated to severe COVID-19. Neuropsychiatric symptoms raise concerns about long-term central nervous system impacts, while studies on COVID-19 vaccines suggest potential improvements in neurological autoimmune disorders.

The intricate interaction between COVID-19 and the central nervous system calls for further research into potential therapeutic interventions. Overall, understanding these complex pathways is crucial, especially for individuals with pre-existing neurodegenerative conditions, to mitigate the long-term consequences of the disease.

## Figures and Tables

**Figure 1 vaccines-12-00222-f001:**
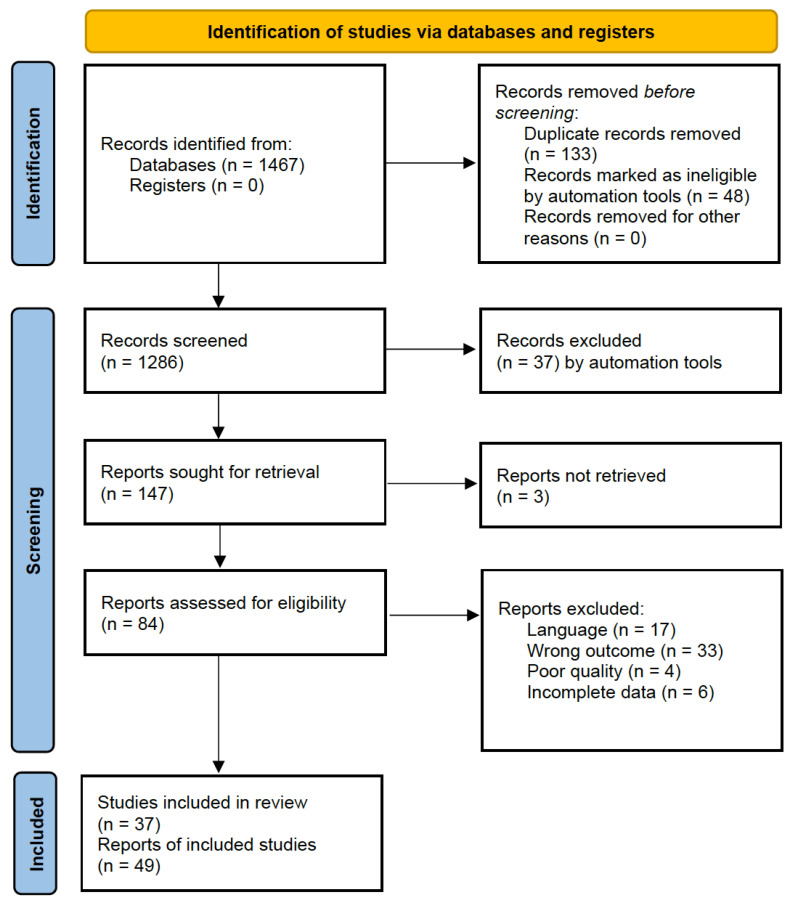
PRISMA flow diagram.

**Table 1 vaccines-12-00222-t001:** Neurobiological effects of COVID-19 in patients.

Evidence Level	Number of Patients	Neurologic Symptoms	Neurological Disease	Mechanism of Entry	Cerebrovascular Disease	Study
Review of available literature	Specific number (not mentioned)	Headache, dizziness, altered mental status, anosmia (loss of smell), ageusia (loss of taste), muscle weakness, seizures	Encephalitis, encephalopathy, stroke, Guillain-Barré syndrome, acute disseminated encephalomyelitis (ADEM), peripheral neuropathy	SARS-CoV-2 can enter the CNS through the olfactory bulb, direct invasion, retrograde axonal transport, or infection of endothelial cells in brain blood vessels	Increased risk of cerebrovascular events, such as ischemic stroke and intracerebral hemorrhage	Ellul et al., 2020 [18]
Observational study (moderate level)	214 hospitalized patients	Dizziness, headache, impaired consciousness, taste impairment, smell impairment, vision impairment, ataxia, seizure, nerve pain	Impaired consciousness, acute cerebrovascular disease, ataxia, seizure, taste impairment, smell impairment, vision impairment, nerve pain	Expression and distribution of ACE2 receptor suggest direct or indirect mechanisms	Acute cerebrovascular disease, including ischemic stroke and cerebral hemorrhage	Mao et al., 2020 [9]
Total of 35 patients	Specific number (not mentioned)	Headache, myalgias, loss of smell and taste, cognitive impairment	Not specifically mentioned	Hematogenous or retrograde neuronal route	Not specifically mentioned	Almeria et al., 2020 [19]
Meta-analysis of 145 papers	Exact number not specified	Altered mental status, headache, dizziness and balance problems, seizures	Stroke, encephalopathy, Guillain-Barré syndrome (GBS), meningitis and encephalitis, neuropathy	Direct invasion through ACE2 receptors, indirect pathway through systemic inflammation and immune responses, neuronal transmission through olfactory nerve, hematogenous spread through crossing of the blood–brain barrier	Ischemic stroke, hemorrhagic stroke, large-vessel occlusion, multi-territory infarcts, cryptogenic stroke	Nannoni et al., 2020 [20]
Autopsies of 33 COVID-19 patients	Total of 33 patients	Impaired consciousness (5 patients), intraventricular hemorrhage (2 patients), headache (2 patients), behavioral changes (2 patients)	Acute infarcts in 6 patients, acute cerebral ischemia in 2 patients	Crossing the neural–mucosal interface in the olfactory mucosa, following neuroanatomical structures in the medulla oblongata	Acute cerebrovascular disease reported in some patients	Meinhardt et al., 2021 [21]
Review of 72 studies	Specific number (not mentioned)	Altered consciousness, encephalopathy, confusion, disturbance of consciousness	Neuroimaging abnormalities, cerebrospinal fluid abnormalities, EEG changes	Via the angiotensin-converting enzyme 2 (ACE2) receptor	Acute cerebrovascular events, such as strokes; possibly multifactorial mechanisms involving direct viral invasion, procoagulant state, hypoxia, and immune response	Rogers et al., 2020 [22]
Retrospective study	Retrospective cohort studies in the USA	Delirium, anxiety, depression, poor memory, insomnia, manic symptoms	Stroke	Believed to enter the brain through direct invasion, hematogenous spread, or neuronal transmission	Ischemic stroke and other cerebrovascular events	Taquet et al., 2021 [23]
Review of available literature	Specific number (not mentioned)	Headache, epilepsy, altered consciousness, encephalitis	Meningitis/encephalitis	Possibly invades the brain through the olfactory tract in the early stages of infection	Not specifically mentioned	Guo et al., 2020 [24]
Retrospective observational registry	Specific number (not mentioned)	Altered mental status, headache, seizure	Acute ischemic stroke, intracranial hemorrhage, cerebral venous sinus thrombosis	SARS-CoV-2 enters host cells through ACE2 receptor, expressed in various human tissues including CNS	Incidence rate of cerebrovascular events in COVID-19 patients higher than that in non-COVID-19 patients	Siegler et al., 2020 [25]

**Table 2 vaccines-12-00222-t002:** Stratification of neurological effects of SARS-CoV-2 by age group.

Age Group	Common Neurological Risks	Neurological Complications	Study
Children	Shared abnormalities in the brain and spine, common CNS manifestations	- Acute disseminated encephalomyelitis - Myelitis - Neural enhancement - Splenial lesions - Myositis	Lindan et al., 2020 [30]
		- Acute encephalopathy or encephalitis - Acute necrotizing encephalopathy - Epilepsy/seizures - Acute transverse myelitis - Guillain-Barré syndrome (GBS) - Posterior reversible encephalopathy syndrome - Acute ischemic or hemorrhagic stroke	LaRovere et al., 2021 [31]
		- Exposure to SARS-CoV-2 during pregnancy is associated with an increased risk of neurodevelopmental disorders in offspring. These disorders primarily involve developmental disorders of motor function or speech and language.	Edlow et al., 2022 [32]
		- Altered awareness, seizures, and headache - Other neurological complications such as encephalitis, seizures, and cerebrovascular infarct have been reported in small series or single case reports. However, these severe neurological manifestations are rare.	Riva et al., 2021 [33]
Adults	Neurological symptoms likely a result of the body’s immune response	- Encephalitis - Cerebrovascular disease - Peripheral nervous system (PNS) manifestations: sensory ailments and neuralgia can occur, affecting the peripheral nerves - Neuromuscular injury - Altered mental state: approximately 1 out of 3 COVID-19 patients may experience changes in mental status, such as confusion and agitation	Kalra et al., 2021 [34]
		- Anosmia (loss of smell) and ageusia (loss of taste): these are often early symptoms of COVID-19 and can occur even in mild cases	Boldrini et al., 2021 [35]
Elderly	Increased risk of neurological manifestations due to aging	- Elderly individuals infected with SARS-CoV-2 are at a higher risk of experiencing neurological complications. Common neurological risks in elderly COVID-19 patients include encephalitis, altered mental status, seizures, confusion, agitation, sensory ailments, neuralgia, acute ischemic stroke, and neuropsychiatric symptoms such as delirium.	Kalra et al., 2021 [34]

## Data Availability

No new data were created in this study.

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
