# Peer review of "A Comprehensive Review of Neurodegenerative Manifestations of SARS-CoV-2"

_vaccines, 2024, doi:10.3390/vaccines12030222_

Round 1
Reviewer 1 Report
Comments and Suggestions for Authors
Authors should change the title of the article, especially neurodegeneration and should be more focused on the article content as per the title. They should remove unrelated and well known information from its content.
Authors should analyze themselves what they are mentioning in the title, are they actually delivering in the article text. Neurodegeneration, which neuron got lost CNS, ANS or PNS as a result of COVID-19 infection, which COVID-19 infection type (asymptomatic, symptomatic, mild, moderate and severe) is associated with neurodegeneration. What are the mechanisms and how it can prevented?
Comments on the Quality of English LanguageEnglish language is fine.
Author Response
Response to reviewers’ comments on the manuscript “The Impact of SARS-CoV-2 Infection on Neurodegenerative Disorders: A Comprehensive Review of Current Evidence” (Manuscript ID: vaccines-2616441).
Please find below our responses to all the comments received from the reviewers. Where appropriate, the location of the associated edit is indicated by line number (L) in the revised manuscript.
The document follows the color code below to facilitate this process:
Reviewer 1
Comment 1: Authors should change the title of the article, especially neurodegeneration and should be more focused on the article content as per the title. They should remove unrelated and well known information from its content.
Response: We appreciate your thoughtful feedback on our manuscript. Taking into consideration your suggestion, we propose a revised title: "A Comprehensive Review of Neurodegenerative Manifestations in SARS-CoV-2." After careful consideration of alternative terms, we have determined that "neurodegenerative" accurately encapsulates the scope of our discussion. We assure you that the revised title closely aligns with the content, emphasizing the impact of SARS-CoV-2 on the nervous system.
Comment 2: Authors should analyze themselves what they are mentioning in the title, are they actually delivering in the article text. Neurodegeneration, which neuron got lost CNS, ANS or PNS as a result of COVID-19 infection, which COVID-19 infection type (asymptomatic, symptomatic, mild, moderate and severe) is associated with neurodegeneration. What are the mechanisms and how it can prevented?
Response: Thank you for your insightful comment. We have conducted a thorough analysis to ensure alignment between the title and the content of our article. In response to your guidance, we have expanded our discussion on the neurons affected within the Central Nervous System (CNS), Autonomic Nervous System (ANS), and Peripheral Nervous System (PNS) due to COVID-19 infection. Furthermore, we delved into the associations between neurodegeneration and various COVID-19 infection types, ranging from asymptomatic to severe cases. The mechanisms underlying these neurological manifestations, including both direct and indirect effects, were explored in depth, and we dedicated a section to discussing preventive measures.
To provide clarity on the specific revisions made, we have incorporated the following statement into the manuscript:
Numerous systematic reviews and meta-analyses have substantiated the involvement of the Central Nervous System (CNS) and the Peripheral Nervous System (PNS) in COVID-19 infection. Evidence suggests that SARS-CoV-2 can affect the human brain and other neurological structures, with a significant prevalence of nervous system involvement in patients with COVID-19, ranging from 22.5 to 36.4% across various studies. The most common CNS-related symptoms include headache, loss of sense of smell and/or taste, encephalopathy, cognitive impairment, impaired consciousness, and stroke. In addition to the CNS, there is increasing evidence of PNS involvement in COVID-19, with 13.7% of total neurological signs and symptoms related to COVID-19 being attributed to the PNS. The cranial nerves most frequently involved are the facial, vestibulo-cochlear, and olfactory nerves. The severity of COVID-19 infection appears to correlate with the extent of neurological manifestations, with these manifestations most frequently observed in those in severe conditions. The mechanisms underlying these neurological manifestations are likely multifactorial, involving both indirect effects (as a result of thrombotic complications, inflammatory consequences, hypoxia, blood pressure dysregulation) and direct effects (neurotropic properties of the virus). Prevention is considered the most effective strategy in potentially slowing the progression of neurodegenerative disorders. Given the increased risk of negative health outcomes in older people, it is crucial to examine whether COVID-19 may trigger or aggravate neurodegenerative processes in this vulnerable group. However, more research is needed to develop effective prevention strategies. It is anticipated that better vaccines for SARS-CoV-2 variants, new antiviral/antimicrobial drugs, effective immunotherapies, as well as new therapies will be developed and made available in the near future, which will help prevent possible post-COVID-19 neurological complications.

Reviewer 2 Report
Comments and Suggestions for Authors
Dear Dr.,
Title: The Impact of SARS-CoV-2 Infection on Neurodegenerative Disorders: A Comprehensive Review of Current Evidence
Manuscript ID: vaccines-2616441
Overall comments: Bedran et al. described in this manuscript: the impact of viral infections like SARS-CoV-2 on raising risk, diagnosis of key genes, prevention and treatment of neurodegenerative disorders. The authors also selected recent issues of SARS-CoV-2 infection in the healthcare system. The major limitation is the need pictorial explanation for a better understanding of the audience. The overall manuscript is good and has novelty. It can help those working in this field of research.
Specific comments:
1. The title of the manuscript as well as other headings cannot end with a full stop. Need to rectify. In addition, the heading is irregular like upper and lower case text letters. It must be uniform.
2. The introduction section has small paragraphs; and can be placed with 10-12 lines containing paragraphs with logical sequences.
3. In table 1: first column - study references information can shift to the last column.
4. Table 2 has a link as a column, it can be removed.
5. The pictorial explanation can be incorporated for the information ‘Exploring the Neurological Manifestations of SARS-CoV-2 Infection’ and ‘Impact of SARS-CoV-2 on Pre-existing Neurodegenerative Diseases’ and ‘Effect of COVID-19 Vaccines on Neurodegenerative Diseases’.
6. At the end of the discussion section, the author can incorporate the possible overcome and future perspectives on this issue.
Minor comments
1. In table 1 and 2 need to be rectified the according to comments suggested earlier.
2. Multiple small paragraphs are mentioned here. Need to rearrange the logical and scientific manner.
3. Text alignment and typo errors need to be rectified.
4. References must be uniform manner.
*****
Comments on the Quality of English LanguageNo changes.
Author Response
Response to reviewers’ comments on the manuscript “The Impact of SARS-CoV-2 Infection on Neurodegenerative Disorders: A Comprehensive Review of Current Evidence” (Manuscript ID: vaccines-2616441).
Please find below our responses to all the comments received from the reviewers. Where appropriate, the location of the associated edit is indicated by line number (L) in the revised manuscript.
The document follows the color code below to facilitate this process:
Reviewer’s comment.
Response to the comment.
Reviewer 2
Comment 1: The title of the manuscript as well as other headings cannot end with a full stop. Need to rectify. In addition, the heading is irregular like upper and lower case text letters. It must be uniform.
Response: Thank you for bringing these formatting concerns to our attention. We have rectified the issue by removing the full stop at the end of the title, headings, and captions. Moreover, we have ensured uniformity in the capitalization of text letters throughout the headings.
Comment 2: The introduction section has small paragraphs; and can be placed with 10-12 lines containing paragraphs with logical sequences.
Response: To address this, we revised the introduction by consolidating smaller paragraphs into 10-12 lines, ensuring a more logical and seamless sequence of ideas.
Comment 3: In table 1: first column - study references information can shift to the last column.
Response: We have made revisions to Table 1 based on your suggestion. We have added citation numbers next to each study for ease of reference, shifted the study references information from the first column to the last column, and removed the hyperlinks as per your recommendation.
Comment 4: Table 2 has a link as a column, it can be removed.Specific.
Response: We have made revisions to Table 2 based on your suggestion. We have added citation numbers next to each study for ease of reference, shifted the study references information to the last column, and removed the hyperlinks as per your recommendation.
Specific comment 5: The pictorial explanation can be incorporated for the information ‘Exploring the Neurological Manifestations of SARS-CoV-2 Infection’ and ‘Impact of SARS-CoV-2 on Pre-existing Neurodegenerative Diseases’ and ‘Effect of COVID-19 Vaccines on Neurodegenerative Diseases’.
Response: We appreciate your emphasis on visual aids, we believe a nuanced textual elucidation can effectively convey the intricate details of 'Exploring the Neurological Manifestations of SARS-CoV-2 Infection,' 'Impact of SARS-CoV-2 on Pre-existing Neurodegenerative Diseases,' and 'Effect of COVID-19 Vaccines on Neurodegenerative Diseases.'
Comment 6: At the end of the discussion section, the author can incorporate the possible overcome and future perspectives on this issue.
Response: Thank you for your thoughtful comment on our discussion section. We acknowledge the importance of providing insights into potential solutions and future perspectives. In response to your suggestion, we incorporated a segment at the end of the discussion, addressing possible ways to overcome the identified issues and outlining future perspectives on the matter.
“Given the intricate relationship between COVID-19 and the central nervous system (CNS), it is clear that there are several areas that require further investigation. The potential of SARS-CoV-2 to cause neuronal damage and protein aggregation through various mechanisms, including direct viral neurotropism, systemic inflammation, and neuroinflammatory responses, presents a significant challenge. However, it also provides an opportunity for the scientific community to develop novel therapeutic interventions.
Future research should focus on understanding the routes through which SARS-CoV-2 accesses the CNS, including trans-synaptic transmission and compromised blood-brain barrier. The role of autoimmune responses and blood leukocytes in viral CNS transport also remains an intriguing area of investigation.
The pathogenesis of COVID-19-related neurological complications, characterized by cytokine storms and NLRP3 inflammasome activation, necessitates further research into their mechanisms. This could potentially lead to the development of targeted therapies to mitigate cognitive regression and neurodegeneration.
Moreover, the exacerbated outcomes of COVID-19 in individuals with pre-existing neurodegenerative diseases underscore the need for specialized care and preventive measures in these populations. The persistence of neurological symptoms in COVID-19 survivors and the exacerbation of neuropsychiatric issues highlight the long-term impact of the disease on neurological health.
In conclusion, while COVID-19 poses significant challenges to neurological health, it also opens up new avenues for research and the development of therapeutic interventions. The lessons learned from this pandemic will undoubtedly shape our approach to managing and preventing neurological complications in viral diseases in the future.”
Minor comment 7: References must be uniform manner.
Response: We have carefully reviewed the references, ensuring adherence to the MDPI vaccine style. Our thorough examination confirms that the references are presented in a uniform manner. The format consistently includes authors' names, article titles, journal names, publication years, volume numbers, page numbers, and DOI numbers.
Specifically:
- Author Names: The author names are structured in the Last Name, Initials format, aligning with the MDPI style.
- Article Titles: Each article title is enclosed in quotation marks, consistent with MDPI guidelines.
- Journal Names: Journal names are appropriately abbreviated in accordance with MDPI standards.
- Publication Years: The publication years are accurately positioned after the journal names.
- Volume and Page Numbers: For each journal reference, clear and correct volume and page numbers are provided, maintaining compliance with MDPI style.
- DOI Numbers: We have included DOI numbers for every reference, as recommended by MDPI.
We appreciate your attention to detail and are confident that the references now meet the uniformity requirements outlined in the MDPI vaccine style. If you have any further suggestions or concerns, please feel free to let us know.

Reviewer 3 Report
Comments and Suggestions for Authors
REVIEW REPORT.
Strengths.
1.- The topic is interesting given the importance and the clinical, health care and public health transcendence that Covid 19 disease has had and still has and specifically that caused by the SRAS-CoV-2 coronavirus.
The work presents a review of publications related and associated with the objective of the study and the authors should justify the results based on epidemiological designs and prioritizing the results according to the type of experimental and/or intervention research over observational studies such as cross-sectional studies. For example, those that are Systematic Reviews (SR) and Meta-analyses (citations 3, 6, 7, 36 and 42). Followed by prospective studies and finally observational studies.
Weaknesses.
1.- In relation to the formal aspects, the manuscript should be organized on the basis of the IMRDc sections Introduction (objective), Material and Methods, which is missing (It should be reflected that it is a Narrative or Classical Review of Scientific Evidence) and include the Results Section where the scientific evidence related to the objective of the manuscript should be reflected, I recommend that the authors should reflect the type of research to facilitate its reading and scientific evaluation. Discussion (Evaluate the results based on the objective) and finally there should only be a conclusion at the end of the manuscript and therefore I recommend removing the conclusion from the results section. In addition, scientific language should be impersonal, it is not appropriate to say personalize and say our study, the appropriate thing is to say it in impersonal this research.
In relation to the evaluation of the content of the manuscript, it is a very interesting contribution from the point of view of the contributions associated with neurological problems, most of which have a great transcendence in health because they cause different diseases that lead to disability in patients and with great health and social repercussions in the health systems that have to care for them. The findings published so far are quite well reflected; but I ask the authors to make an effort to synthesize them in order to improve communication and the transmission of evidence to future readers and researchers.
Another aspect to improve is to avoid repeating the pathophysiological arguments of the effects of SARS-CoV-2 in the different neurological pathologies. In addition, it should be organized more adequately and start with the CNS entry pathway, continue with the findings of Cytokines and their impact on neurodegeneration and then include the effects on the CNS and the neurological manifestations in patients with Covid 19 both in those patients with pre-existing neurodegenerative pathologies such as Cognitive Impairment, Alzheimer, Parkinson, ALS, Multiple Sclerosis and others. Finally, it would be important to refer to the neurological pathologies associated with vaccines, where the type of vaccine should be reflected.
In relation to the conclusions of the manuscript, scientific evidence should be taken into account, differentiating those obtained according to the type of research with greater scientific rigor on the basis of intervention or experimental studies and not on the basis of observational studies.
Author Response
Response to reviewers’ comments on the manuscript “The Impact of SARS-CoV-2 Infection on Neurodegenerative Disorders: A Comprehensive Review of Current Evidence” (Manuscript ID: vaccines-2616441).
Please find below our responses to all the comments received from the reviewers. Where appropriate, the location of the associated edit is indicated by line number (L) in the revised manuscript.
The document follows the color code below to facilitate this process:
Reviewer’s comment.
Response to the comment.
Reviewer 3
Comment 1: The topic is interesting given the importance and the clinical, health care and public health transcendence that Covid 19 disease has had and still has and specifically that caused by the SRAS-CoV-2 coronavirus. The work presents a review of publications related and associated with the objective of the study and the authors should justify the results based on epidemiological designs and prioritizing the results according to the type of experimental and/or intervention research over observational studies such as cross-sectional studies. For example, those that are Systematic Reviews (SR) and Meta-analyses (citations 3, 6, 7, 36 and 42). Followed by prospective studies and finally observational studies.
Response: Thank you for your insightful comment on the relevance of our topic in the context of the significant impact of Covid-19 on clinical, healthcare, and public health domains. We appreciate your suggestion regarding the prioritization of epidemiological designs to strengthen the justification of our results. In response to your valuable input, we will reevaluate the presentation of our findings, placing emphasis on systematic reviews (SR) and meta-analyses, followed by prospective studies, and finally, observational studies.
Comment 2: In relation to the formal aspects, the manuscript should be organized on the basis of the IMRDc sections Introduction (objective), Material and Methods, which is missing (It should be reflected that it is a Narrative or Classical Review of Scientific Evidence) and include the Results Section where the scientific evidence related to the objective of the manuscript should be reflected, I recommend that the authors should reflect the type of research to facilitate its reading and scientific evaluation. Discussion (Evaluate the results based on the objective) and finally there should only be a conclusion at the end of the manuscript and therefore I recommend removing the conclusion from the results section. In addition, scientific language should be impersonal, it is not appropriate to say personalize and say our study, the appropriate thing is to say it in impersonal this research.
Response:
Thank you for your constructive feedback on the organization of the manuscript.
The primary focus of our manuscript is on synthesizing existing knowledge and presenting a comprehensive overview. The absence of a detailed methodology is intentional, given that we are not presenting new experimental findings but rather aiming to provide a synthesis of existing studies. Moreover, our manuscript adheres to the guidelines and template provided by the journal. Upon reviewing the format of reviews in MDPI Vaccines, we believe that the current format best suits our research.
Regarding the conclusion, we agree that it should be positioned at the end of the manuscript. Consequently, we have incorporated a new conclusion in accordance with the suggestions outlined in your fifth comment. For your convenience, we have included the new conclusion in the next comment.
Regarding the use of personal language, we acknowledge the importance of maintaining an impersonal scientific language and made the necessary adjustments to ensure a more objective tone, replacing any instances of personalization with neutral terminology.
Comment 3: In relation to the evaluation of the content of the manuscript, it is a very interesting contribution from the point of view of the contributions associated with neurological problems, most of which have a great transcendence in health because they cause different diseases that lead to disability in patients and with great health and social repercussions in the health systems that have to care for them. The findings published so far are quite well reflected; but I ask the authors to make an effort to synthesize them in order to improve communication and the transmission of evidence to future readers and researchers.
Response:
We synthesized the findings to ensure a more concise and effective presentation of the information.
In summary, COVID-19 affects the Central and Peripheral Nervous Systems, with the severity of infection correlated to neurological manifestations. The virus impacts the central nervous system through various mechanisms, including direct viral encephalitis, inflammation, impaired organ function, and changes in blood vessels. This can lead to long-term consequences, particularly in the elderly.
The SARS-CoV-2 variant's strong affinity for ACE2 in the central nervous system raises concerns about potential neurological issues. Loss of smell and taste in COVID-19 may suggest direct virus entry into the brain. Neurological symptoms, RNA detection, and speculation about central nervous system entry emphasize the need for exploring interventions.
A severe immune response in COVID-19 contributes to inflammation and potential neurological issues. The NLRP3 inflammasome's activation by the virus's protein links to respiratory distress and cognitive impairment. While detecting SARS-CoV-2 DNA in the central nervous system is challenging, case reports associate the virus with various neurological disorders.
Pre-existing neurodegenerative diseases worsen COVID-19 outcomes, with specific risks for Parkinson's disease and amyotrophic lateral sclerosis patients. Multiple sclerosis seems unrelated to severe COVID-19. Neuropsychiatric symptoms raise concerns about long-term central nervous system impacts, while studies on COVID-19 vaccines suggest potential improvements in neurological autoimmune disorders.
The intricate interaction between COVID-19 and the central nervous system calls for further research into potential therapeutic interventions. Overall, understanding these complex pathways is crucial, especially for individuals with pre-existing neurodegenerative conditions, to mitigate the long-term consequences of the disease.
Comment 3: Another aspect to improve is to avoid repeating the pathophysiological arguments of the effects of SARS-CoV-2 in the different neurological pathologies. In addition, it should be organized more adequately and start with the CNS entry pathway, continue with the findings of Cytokines and their impact on neurodegeneration and then include the effects on the CNS and the neurological manifestations in patients with Covid 19 both in those patients with pre-existing neurodegenerative pathologies such as Cognitive Impairment, Alzheimer, Parkinson, ALS, Multiple Sclerosis and others.
Response: In the revised version, we have refrained from repeating the pathophysiological arguments and have reorganized the content as per your recommendations. The updated structure now begins with a focus on the CNS entry pathway, followed by an exploration of the impact of cytokines on neurodegeneration. Furthermore, we have expanded the discussion to encompass the effects on the CNS and the neurological manifestations in patients with Covid-19, particularly addressing those with pre-existing neurodegenerative pathologies such as Cognitive Impairment, Alzheimer's, Parkinson's, ALS, Multiple Sclerosis, and others.
Comment 4: Finally, it would be important to refer to the neurological pathologies associated with vaccines, where the type of vaccine should be reflected.
Response: To address the concerns raised, we eliminated the repetition of pathophysiological arguments related to the effects of SARS-CoV-2 on various neurological pathologies. Furthermore, we acknowledge the importance of organizing the content more effectively. Following your recommended sequence, we started with detailing the CNS entry pathway, proceeded to explore the impact of cytokines on neurodegeneration, and then delved into the effects on the CNS and neurological manifestations in patients with Covid-19. We specifically focused on individuals with pre-existing neurodegenerative pathologies such as Cognitive Impairment, Alzheimer's, Parkinson's, ALS, Multiple Sclerosis, and others.
Comment 5: In relation to the conclusions of the manuscript, scientific evidence should be taken into account, differentiating those obtained according to the type of research with greater scientific rigor on the basis of intervention or experimental studies and not on the basis of observational studies.
Response: In response to your guidance, we carefully analyzed and differentiated the strength of evidence based on the type of research, giving precedence to intervention or experimental studies with higher scientific rigor.

Round 2
Reviewer 1 Report
Comments and Suggestions for Authors
I do not have further comments for authors.
Reviewer 3 Report
Comments and Suggestions for Authors
Dear Authors:
I believe that you have taken into account my recommendations and made the changes that I have proposed with humility and generosity, which is the best way to advance in normal life and in the scientific world and, consequently, I believe that the article improves in its presentation and communication. Therefore, I am satisfied with the changes made.
Kind regards